# Identification of a Putative CodY Regulon in the Gram-Negative Phylum Synergistetes

**DOI:** 10.3390/ijms23147911

**Published:** 2022-07-18

**Authors:** Jianing Geng, Sainan Luo, Hui-Ru Shieh, Hsing-Yi Wang, Songnian Hu, Yi-Ywan M. Chen

**Affiliations:** 1State Key Laboratory of Microbial Resources, Institute of Microbiology, Chinese Academy of Sciences, Beijing 100101, China; gengjn@im.ac.cn (J.G.); luosainan9606@163.com (S.L.); 2University of Chinese Academy of Sciences, Beijing 100049, China; 3Department of Microbiology and Immunology, College of Medicine, Chang Gung University, Taoyuan 333, Taiwan; cowcow119@hotmail.com (H.-R.S.); kenny200929@gmail.com (H.-Y.W.); 4Graduate Institute of Biomedical Sciences, College of Medicine, Chang Gung University, Taoyuan 333, Taiwan; 5Molecular Infectious Disease Research Center, Chang Gung Memorial Hospital, Taoyuan 333, Taiwan

**Keywords:** Synergistetes, CodY orthologues, binding consensus of Synergistetes CodY, CodY regulon of Synergistetes

## Abstract

CodY is a dominant regulator in low G + C, Gram-positive Firmicutes that governs the regulation of various metabolic pathways and cellular processes. By using various bioinformatics analyses and DNA affinity precipitation assay (DAPA), this study confirmed the presence of CodY orthologues and corresponding regulons in Gram-negative Synergistetes. A novel palindromic sequence consisting of AT-rich arms separated by a spacer region of variable length and sequence was identified in the promoters of the putative *codY*-containing operons in Synergistetes. The consensus sequence from genera *Synergistes* and *Cloacibacillus* (5′-AATTTTCTTAAAATTTCSCTTGATATTTACAATTTT) contained three AT-rich regions, resulting in two palindromic sequences; one of which is identical to Firmicutes CodY box (5′-AATTTTCWGAAAATT). The function of the consensus sequence was tested by using a recombinant CodY protein (His-CodY_DSM_) of *Cloacibacillus evryensis* DSM19522 in DAPA. Mutations in the central AT-rich sequence reduced significantly the binding of His-CodY_DSM_, whereas mutations in the 5′ or 3′ end AT-rich sequence slightly reduced the binding, indicating that CodY_DSM_ could recognize both palindromic sequences. The proposed binding sequences were found in the promoters of multiple genes involved in amino acids biosynthesis, metabolism, regulation, and stress responses in Synergistetes. Thus, a CodY-like protein from Synergistetes may function similarly to Firmicutes CodY.

## 1. Introduction

Synergistetes was formally recognized as a novel bacterial phylum based on the 16S rRNA gene phylogeny tree of the bacterial domain in 2009 [1]. The members of the phylum Synergistetes are characterized as Gram-negative, anaerobic bacteria with various cell shapes [1,2]. Microbes of the phylum Synergistetes can utilize amino acids as an energy source and degrade toxic environmental compounds, sludges, and glucose for the production of renewable energy and biogas [1,3,4]. For example, the representative species *Synergistes jonesii*, isolated from goat rumen, can degrade 3-hydroxy-4(1H)-pyridone (3,4-DHP) for growth [5]. Currently, 16 genera are classified in the phylum Synergistetes: *Acetomicrobium*, *Aminiphilus*, *Aminivibrio*, *Aminobacterium*, *Aminomonas*, *Cloacibacillus*, *Dethiosulfovibrio*, *Fretibacterium*, *Jonquetella*, *Lactivibrio*, *Pacaella*, *Pyramidobacter*, *Rarimicrobium*, *Synergistes*, *Thermanaerovibrio* and *Thermovirga* [2,6,7,8,9,10]. Microbes of the phylum Synergistetes reside in extended anaerobic environments, including animal and insect gastrointestinal tracts, animal wastes, wastewater treatment systems, soils, and oil wells. Moreover, several species of Synergistetes are present in the oral microbiome and are thought to be associated with periodontal diseases [11,12].

The genome sequences of 35 cultivable strains of Synergistetes, 286,997 pieces of metagenomes from the human gut microbiome, and 2870 pieces of metagenomes from various environmental microbiomes, including the metagenomes of samples from activated sludge, anaerobic digesters, and groundwater are currently available in the NCBI Entrez genomes database [13,14,15,16]. The availability of these sequences provides opportunities for the identification of global regulators that are present in bacterial groups at different taxonomic levels using bioinformatic analyses. Genomic analyses indicate that the genomes of Synergistetes are tightly arranged, with an average of approximately 1000 genes per megabase. While a few species are asaccharolytic, all members of the phylum Synergistetes are able to ferment amino acids [2]. The average percentage of genes encoding proteins associated with amino acid uptake and metabolism in a Synergistetes genome is 11.8 ± 0.8%, representing the highest proportion among all bacterial phyla to date [17].

CodY was first discovered in *Bacillus subtilis* in 1995 as a repressor of the dipeptide permease operon (*dppABCDE*) [18]. Later studies confirmed that CodY is a master regulator for various metabolic genes, and that regulation by CodY facilitates the adaptation of bacteria to poor nutritional conditions [19]. Subsequent studies confirmed that CodY is widely distributed and highly conserved among low G + C, Gram-positive bacteria, including pathogens such as *Streptococcus pneumoniae* and *Listeria monocytogenes* [20,21,22], in which CodY also regulates the expression of multiple virulence genes [23]. CodY is composed of an N-terminal GAF (cGMP-stimulated phosphodiesterase, adenylate cyclase and a bacterial transcription regulator FhlA) domain and a C-terminal winged helix-turn-helix (wHTH) DNA-binding domain. The GAF domain responds to stimuli and binds with branched-chain amino acids (BCAA, e.g., isoleucine, leucine, and valine) or GTP to activate the DNA-binding activity of CodY [24,25]. Upon activation, the wHTH domain of CodY binds to a palindromic DNA sequence (5′-AATTTTCWGAAAATT) to repress the expression of the downstream gene [20,26,27]. Although GAF and wHTH domains have been described in many proteins, CodY is the only known protein that harbors both a GAF domain and a wHTH domain, and, thus, it is possible to search for putative CodY in bacteria of all phyla using this characteristic, including phyla that are evolutionarily distant from Firmicutes.

In this study, a GAF-wHTH containing, CodY-like protein and its binding consensus sequence were identified in species of the phylum Synergistetes. Additionally, the interaction between the CodY-like protein and the predicted binding sequence was demonstrated using a DNA-affinity precipitation assay (DAPA), indicating that microbes of the Synergistetes phylum likely harbor a CodY-like global regulatory protein for regulation of multiple cellular activities.

## 2. Results

### 2.1. Identification and Characterization of CodY-like Proteins in the Phylum Synergistetes

As CodY is the only known protein composed of a GAF and a wHTH domain, profile hidden Markov models (HMMs) were used to investigate whether a CodY-like protein, i.e., proteins with the same domain composition, existed outside of the phylum Firmicutes. An interrogation of bacterial genome sequences in the NCBI Reference Sequence Database (RefSeq), ecological environment metagenomes in the NCBI Entrez genomes database, and human gut microbiome datasets in the European Bioinformatics Institute database (EBI), unexpectedly revealed 107 loci encoding proteins containing both domains in species of the phylum Synergistetes. Among these, 35 loci were present in the genomes of 20 cultivable species, across 12 genera (Table 1), and 23 loci were found in the human gut microbiome datasets across four genera (Appendix A). The remaining 49 loci were present in different contings obtained from ecological samples of extreme environments, such as wastewater treatment lagoons, anaerobic digester sludge, activated sludge enrichment, groundwater enrichment, hot springs enrichment, sulfur mats, and the deep marine trench. Thus, a CodY-like protein is widely distributed in the phylum Synergistetes, including species occupying different ecological niches.

To analyze the evolutionary relationship of the newly identified CodY-like protein and CodY of Firmicutes, we first analyzed the phylogenic relationship of all CodY proteins identified in Firmicutes (data not shown), and randomly selected one locus from each evolution branch, resulting in a total of 17 loci. The CodY-like proteins of the 35 cultivable isolates of the phylum Synergistetes were then compared with the CodY proteins of the 17 representative species of the phylum Firmicutes (Appendix A) at the deduced amino acid sequence level (Figure 1). Protein identity cluster analysis revealed two main groups: one containing the CodY proteins of Firmicutes, and the other containing the CodY-like proteins of Synergistetes. Thus, CodY and the CodY-like protein of species in the same phylum are more homologous to each other and the extent of similarity decreased across phyla. The putative CodY protein of the phylum Synergistetes displays an average of 43.32% identity (35.32% to 50.20%) with the CodY protein of Firmicutes, while an average of 70.23% and 55.48% identity are observed between species of the phyla Synergistetes and Firmicutes, respectively. The high identity between species of Synergistetes (70.23%), compared to that of Firmicutes (55.48%), is likely due to the relatively low number of species in Synergistetes.

### 2.2. The Origin of the CodY-like Protein of the Phylum Synergistetes

To examine whether the phylogenetic relationship of CodY and the CodY-like protein follow the same evolutionary history of the diversification of bacterial species, we constructed a 16S rRNA gene-based phylogenetic tree with 51 members of 7 phyla, including first, 5 phyla that do not possess CodY (Appendix A): Proteobacteria, Fusobacteria, Bacteroidetes, Actinobacteria and Aquificae; second, the CodY-possessing Firmicutes; and third, Synergistetes. As shown in Figure 2A, species of the phyla Synergistetes and Firmicutes are located in two independent clades. An identical evolutionary relationship was found with regards to the CodY and CodY-like protein (Figure 2B), where the two proteins are also located in two independent branches. Furthermore, in both phylogenic trees, three major groups were observed under the Synergistetes clade, indicating that the evolution of the CodY-like protein follows the evolution of the bacterial species.

### 2.3. Analysis of the GAF-wHTH Domain

Although only approximately 40% identity was detected between CodY and the CodY-like protein, it is interesting to note that the amino acid sequence of the wHTH domain region of these proteins is relatively conserved across phyla, with an average 73.40% identity (Appendix A). All conserved amino acids within the key DNA-binding region (amino acids 203–222) of CodY [28] are also present in the CodY-like proteins, except for Ile212, which is a Val residue in the CodY-like proteins. The conservation of this region suggests that the CodY of Firmicutes and the CodY-like protein of Synergistetes could recognize a similar DNA sequence. On the other hand, the GAF domains of CodY and the CodY-like protein only share an average of 33.41% identity. Nevertheless, the conserved Arg8, Gln15, Glu144 and Thr148 that are essential for the dimerization of CodY in *B. subtilis* [29] are all present in the corresponding position in the CodY-like protein of Synergistetes, suggesting that the CodY-like protein also acts as a dimer. Although the precise binding motif of BCAA to the GAF domain of the Firmicutes CodY is not well-characterized, studies have shown that amino acid residues Phe40, Arg61, Met62, Met65, Phe71, Pro72, Tyr75, Phe98 and Pro99 are essential for BCAA binding in *B. subtilis* [25,30]. However, the amino acid residues found in the corresponding positions of the CodY-like proteins were dissimilar to this set of essential residues. Furthermore, the key residues Phe40 and Phe71 are absent in all CodY-like proteins, whereas Phe98 is present only in the CodY-like proteins found in the genera *Jonquetella*, *Pyramidobacter*, *Dethiosulfovibrio*, *Aminobacterium*, *Aminivibrio* and *Fretibacterium*. Thus, the effectors for activation of the DNA-binding activity of the CodY-like protein of Synergistetes may differ from that of the CodY from Firmicutes.

### 2.4. The Binding Consensus of the CodY-like Protein

The strategy to define the binding consensus of the CodY-like protein was made based on two observations. Firstly, the similarity in the wHTH domain between CodY and the CodY-like protein would suggest that these two proteins recognize similar sequence. Secondly, the expression of *codY* is autoregulated in microbes of Firmicutes. Specifically, the CodY box is present in the promoters of *codY* of species in the genera of *Lactococcus*, *Streptococcus*, *Enterococcus*, *Staphylococcus* and *Clostridium* [20]. Furthermore, the autoregulation by CodY has been demonstrated experimentally in *Lactococcus lactis* [26], *S. pneumoniae* [21], and *Streptococcus pyogenes* [31]. Thus, it is hypothesized that the expression of the putative *codY* is autoregulated in Synergistetes and it is feasible to identify the binding consensus of the CodY-like protein in the promoters of the putative *codY*. Based on the distance between the putative *codY* and its 5′ flanking loci in the genomes of Synergistetes and results from operon analysis by operon-mapper [32], the putative *codY* is likely to be a part of an operon, with *xerC* being the first gene of the operon. Thus, an unbiased search for the canonical CodY box was performed in the promoter regions of the putative *codY* operon of 35 cultivable strains in the phylum Synergistetes. A sequence of 5′-AATTTTCTTAAAATT, which is identical to the CodY box of Firmicutes, was found in the promoter regions of 12 isolates of the genera *Synergistes* and *Cloacibacillus*, but not in other genera. To verify whether additional consensus sequences could be recognized by the CodY-like protein, we extracted the 5′-flanking regions of *xerC* from genomes of cultivatable isolates, human gut metagenome-assembled genomes, and ecological metagenome assembly sequences in the phylum Synergistetes, and used the MEME motif-searching algorithm [33] to search for conserved palindromic sequences in each genus. As shown in Appendix A and Figure 3, four consensus sequences were discovered in the genera *Synergistes*/*Cloacibacillus*, *Thermanaerovibrio*, *Pyramidobacter*, and *Acetomicrobium*, respectively. No consensus motifs were discovered in other genera because less than three genomes were available for a specific genus. The consensus sequence identified in the genera *Synergistes* and *Cloacibacillus*, 5′-AATTTTCTTAAAATTTCSCTTGATATTTACAATTTT, contains a Firmicutes CodY box (underlined) and an AAAATT/AATTTT inverted repeat (double underlined). Upon close examination of the consensus found in the genera *Thermanaerovibrio* (5′-AATATT-N_21_-AAAATA), *Pyramidobacter* (5′-ATTTTY-N_14_-AAAAAT), and *Acetomicrobium* (5′-WWTTTT-N_12_-ATAATT), it was noted that all of the sequences contained a pair of AT-rich palindromic arms separated by a spacer region of variable length and sequence, suggesting that the binding sequence of the CodY-like protein of the phylum Synergistetes is a palindromic sequence consisting of two AT-rich arms and a variable spacer region.

### 2.5. The CodY-like Protein of Cloacibacillus Evryensis DSM19522 Could Interact with Both the Canonical CodY Box and an AAAATT-N_15_-AATTTT Palindromic Sequence

As the consensus sequence identified in the genera *Synergistes* and *Cloacibacillus* contains both a Firmicutes CodY box and an extended palindromic sequence, the binding specificity of the CodY-like protein of *C. evryensis* DSM19522 (CodY_DSM_) was analyzed using recombinant histidine-tagged CodY_DSM_ (His-CodY_DSM_) in DAPA experiments, as detailed in Materials and Methods. His-CodY_DSM_ was readily induced and purified from the recombinant *E. coli* strain (Appendix A), but the solubility of this protein was rather low (0.2–0.4 μg mL^−1^). The predicted binding sequence at the 5′ flanking region of *xerC* (CLOEV_RS08635) was extracted from the genome of *C. evryensis* DSM19522 (NZ_KK073872.1) and used to design probes for the DAPA experiment (Appendix A). The DAPA results indicated that His-CodY_DSM_ could interact specifically with the full-length probe in a dose-dependent manner (Figure 4). When the three AT-rich sequence were substituted with GGGGGG or six random bases, the signals were abolished, indicating that AT-rich palindromic sequences are critical for the binding of the CodY-like protein. While substitutions of the 5′ end AATTTT with CCCCCC reduced marginally the binding of His-CodY_DSM_, mutations in the 3′ AATTTT did not affect the binding of His-CodY_DSM_ significantly, indicating that His-CodY_DSM_ could interact with both the canonical CodY box and an AAAATT-N_15_-AATTTT inverted repeat sequence, and possibly with a slightly higher affinity to the canonical CodY box. On the other hand, mutations of the center AAAATT significantly reduced the binding, confirming that the CodY-like protein could interact with both targets. Thus, the obtained consensus sequence contains two CodY_DSM_ binding sites; one of which is identical to the CodY box of Firmicutes, and the other is AATTTT-N_15_-AAAATT.

### 2.6. Identification and Functional Prediction of the CodY-like Regulon in the Phylum Synergistetes

As the first step to investigate whether the CodY-like protein also acts as a global regulator in Synergistetes, a search for the proposed binding sequence of the CodY-like protein was performed in the intergenic regions of the 21 complete genomes of Synergistetes, which includes genera *Synergistetes*, *Cloacibacillus*, *Pyramidobacter* and *Acetomicrobium*. The genus *Thermanaerovibrio* was not included as only two complete genomes are available (Table 1). To be noted, since the DAPA analysis revealed that His-CodY_DSM_ could interact with both the CodY box of Firmicutes (5′-AATTTTTCWGAAAATT) and a palindromic sequence with an extended spacer region (Figure 3 and Appendix A), both motifs were used to search for potential targets in the corresponding genus. It was found that the consensus sequence was widely distributed in the intergenic regions of genomes analyzed (Appendix A), accounting for approximately 8–13% of the genes in a genome (Figure 5 and Appendix A). Although the percentages were comparable between the four genera, genera *Synergistetes* and *Cloacibacillus* showed higher numbers of loci that harbor both motifs compared to those of genera *Acetomicrobium* and *Pyramidobacter*. The wide distribution of this novel consensus sequence in a genome suggests that the CodY-like protein is a global regulator. Among these loci, homologs of 32 loci are parts of the CodY regulon in Firmicutes (Appendix A). These loci include genes encoding proteins involved in BCAA biosynthesis (*ilvA*, *ilvN*, *ilvB*, *ilvD*, *ilvC**, leuC*, WP_051682752.1, WP_200778737.1 and WP_037975652.1), BCAA uptake (WP_037976683.1, WP_037975074.1, WP_037976889.1 and WP_009202431.1), proteolysis (*clpB*, *clpX, clpP, fstH* and *resP*), peptide degradation (WP_037974402.1, *pepV*, *def,* WP_051682620.1, WP_051682707.1, WP_078015053.1, WP_037973903.1, WP_111089616.1, WP_037974668.1 and WP_008708838.1), and peptide and amino acid transport (WP_009201625.1, WP_078015996.1, WP_008709275.1 and WP_120373146.1). This observation supports the hypothesis that the CodY-like global regulator of Synergistetes functions in parallel to Firmicutes CodY. 

To further verify this hypothesis, we computed the functional enrichment of groups of genes listed in Appendix A in the genera *Acetomicrobium*, *Cloacibacillus*, *Pyramidobacter*, and *Synergistes* (Figure 6). Three main clusters emerged, with proteins in the categories of amino acid transport and metabolism, transporters, transcriptional regulators, carbohydrate transport and metabolism, and energy production and conversion being the most conserved in the four genera. The identification of genes encoding transcriptional regulators, including HrcA and regulators of the P-II family, GntR family, and CtsR family (Appendix A), suggests that the regulon of the CodY-like protein would contain a set of genes indirectly regulated by the CodY-like protein. Specifically, regulators of both the PII family and the GntR family are closely associated with the regulation of nitrogen metabolism and the nitrogen-carbon interconversion in both Gram-positive and Gram-negative bacteria. HrcA, a heat-inducible transcription repressor, and CtsR, a transcriptional regulator of class III stress genes [34,35], could be involved in the stress responses. Additionally, transcriptional regulators of the LacI, LysR, TetR/AcrR, MurR/RpiR, MerR and MarR families were also found. These regulators are implicated in the regulation of many cellular processes, including metabolism, metal ion homeostasis, antibiotics resistance, virulence expression and stress responses, and thus, a regulon governed by the CodY-like protein is likely to indirectly control multiple cellular activities in Synergistetes. 

Although the genes in the second cluster were generally less conserved, compared to the genes described above between the four genera, significant conservation within a genus was detected in genes encoding proteins involved in genetic information storage and processing, including DNA replication, recombination and repair, transcription, translation, ribosomal structure and biogenesis, nucleotide transport and metabolism. Genes encoding proteins involved in the biosynthesis of cell envelopes, secondary metabolites biosynthesis, metabolism of cofactors and vitamins, heat-shock proteins and molecular chaperones, and horizontal gene transfer were also present in the second cluster (Figure 6). Loci that were least conserved between genera were genes involved in the metabolism of terpenoids and polyketides, phage production, post-translational modification and protein turnover, the toxin-antitoxin system, cell motility and chemotaxis, CRISPR and other defense systems, and cellular community. Nevertheless, conservation among species within a genus was evident. For instance, loci associated with post-translational modification, protein turnover, cell cycle control, cell division, chromosome partition, coenzyme transport and metabolism, cell motility and chemotaxis, and the CRISPR system were enriched in the genus *Acetomicrobium*, and loci encoding proteins of the toxin-antitoxin systems, lipid transport and metabolism, and other defense systems were enriched in the genus *Synergistes*. Taken together, the loci involved in nutrient uptake, general metabolism and regulation were most conserved between the four genera, whereas loci involved in defense systems and stress responses were less conserved. The variations in the putative regulons of the CodY-like regulatory protein between genera may be associated with the natural niches of these microbes.

## 3. Discussion

CodY has been extensively studied in the Firmicutes for its dominant role in metabolism, stress responses and pathogenesis [23,36,37,38]. This study confirmed the presence of CodY orthologues in Synergistetes, which generally exhibits a G + C content close to, or greater than, 50% [39,40,41,42,43]. Thus, the distribution of CodY and the CodY-like protein in bacteria may be more widespread than previously known, and regulation by CodY and its homologs may have similar impacts on bacteria occupying different niches. The differences in the G + C content, the ecological niches and metabolic activities between Firmicutes and Synergistetes would suggest that CodY and the CodY-like protein evolved divergently. As the evolutionary relationship of these proteins follows the phylogenetic relationship of the genera in phyla Firmicutes and Synergistetes, genes encoding CodY and the CodY-like protein should originate from a common ancestor. The results of this study indicate that these two proteins are orthologues that are conserved in domain structures and functions, but distinct in their overall amino acid sequence. 

Past transcriptomic studies of the CodY regulon in Firmicutes, either by microarray or RNA sequencing, often identified loci without a canonical CodY box in the promoter region. For instance, approximately 60% and 30% of the identified CodY targets in *B. subtilis* and *L. lactis*, respectively, are without a canonical CodY box [20]. Although some of these loci are regulated by transcriptional regulators whose activity is governed by CodY, CodY likely directly regulates the rest of the targets through binding to a modified consensus motif. This hypothesis is supported by more recent studies in *B. subtilis* [44], *L. monocytogenes* [22], and *Staphylococcus aureus* [45], in which a modified CodY-binding consensus generated from two overlapping canonical CodY boxes (5′-AATTTTCWGAAWWTTCWGAAAATT) was reported. Unlike the CodY box described in Firmicutes, the binding consensus in Synergistetes is more like a palindromic sequence with AT-rich arms separated by an extended spacer region, as neither the lengths nor the sequences of the identified consensus sequences fit an overlapping arrangement. Although it is possible that the targets identified in this study do not cover the full regulon, the presence of the binding consensus at the 5′ region of multiple genes strongly suggests that similar to CodY in Firmicutes, this newly identified GAF-wHTH protein is a global regulatory protein in Synergistetes.

As a system for genetic manipulation has yet to be established in microbes of the phylum Synergistetes, we selected the DAPA analysis to analyze the DNA-binding activity of the CodY-like protein. Attempts were also made to use gel electrophoresis mobility shift assays (EMSA) [46] to investigate the binding specificity of the CodY-like protein; however, the solubilities of the recombinant CodY-like proteins of Synergistetes were generally lower than those required for effective analysis by EMSA with biotin-labeled probes. Using Protein-Sol (https://protein-sol.manchester.ac.uk/, accessed on 14 January 2022), we found that the solubility of the recombinant CodY-like protein of Synergistetes was significantly lower than CodY from *Streptococcus salivarius* 57.I [47]. Proteins with a Protein-Sol calculated value above 0.45 are considered highly soluble [48]; and, while the Protein-Sol value of CodY from *S. salivarius* 57.I is 0.719, and the value for the CodY-like protein in *C. evryensis* DSM19522 is 0.418. Similar results were also found in CodY-like proteins of two randomly selected species: *Thermanaerovibrio acidaminovorans* DSM 6589, with a value of 0.455, and *C. porcorum* CL-84, with a value of 0.426. By using Ni-NTA magnetic beads, we were able to perform DAPA experiments in a relatively large reaction volume, compared to EMSA analysis, which circumvented any limitations of the low solubility of the CodY-like protein.

Although the difference in the GAF domain between CodY and the CodY-like protein could result in a difference in the effectors, both proteins are involved in the regulation of BCAA biosynthesis. Thus, BCAA is also likely to be the effector of the CodY-like protein. On the other hand, while both BCAA and GTP are necessary to activate CodY in *B. subtilis* [49], CodY of *L. lactis* [50], *Streptococcus mutans* [51], and *S. pneumoniae* [21] is not activated by GTP, suggesting that the choice and efficiency of effector molecules to activate CodY activity could be affected by the ecological niche and physiologic activity of the bacteria. As microbes of Synergistetes could metabolize a variety of substrates, including DHP [5], plant toxin fluoroacetate [52], and mucin, it is also possible that microbes of Synergistetes could utilize different metabolites to signal intracellular nutrient status, and perhaps to activate the CodY-like protein.

As members of phylum Synergistetes inhabit a wide range of extreme environments, it is expected that they have evolved sophisticated features to cope with environmental stresses, including extreme temperature, pH, toxic compounds and challenges from other microbes or bacteriophages. As loci in motility/chemotaxis, defense mechanisms and CRISPR systems all are less conserved among the regulons governed by the CodY-like proteins in Synergistetes, it would suggest that the regulons could vary according to the natural niches of all species in the phylum. For instance, based on our analysis, the CodY-like protein could potentially regulate the chemotaxis and motility of flagella-bearing *Acetomicrobium thermoterrenum* strain DSM 13490 by regulating the expression of loci encoding the methyl-accepting chemotaxis protein (Mcp) and proteins involved in flagella biosynthesis Thus, regulation by the CodY-like protein could potentially fine-tune the activity of chemotaxis and swimming activity. Such regulation seems necessary for bacteria that reside in constantly changing environments, and the regulation could provide certain advantages for these bacteria to thrive in harsh environments.

Taken together, although Synergistetes and Firmicutes are phylogenetically distinct, both CodY and the CodY-like protein are conserved with respect to their regulatory function. Based on the potential targets of the CodY-like protein, the regulation spectrum of the CodY-like protein overlaps with that of the Firmicutes CodY regulon. Thus, similar to Firmicutes, regulation by the CodY-like protein is likely to be essential for the microbes of Synergistetes to cope with stresses in their natural niches.

## 4. Materials and Methods

### 4.1. Bacterial Genome and Metagenome Database

The cultivable bacterial genomes were obtained from the NCBI Genbank database of sequenced bacteria (last updated August 2020) (http://ftp.ncbi.nlm.nih.gov/refseq/release/, accessed on 24 August 2020). Metagenomic data used in this study included genomes obtained from samples from various ecological sources (https://www.ncbi.nlm.nih.gov/Taxonomy/Browser/wwwtax.cgi?mode=Undef&id=408169&lvl=3&keep=1&srchmode=1&unlock/, accessed on 26 Octomber 2020), and human gut metagenomic data (http://ftp.ebi.ac.uk/pub/databases/metagenomics/mgnify_genomes/human-gut/v1.0/uhgp_catalogue/uhgp-100.tar.gz/, accessed on 6 November 2020).

### 4.2. Identification of CodY Homologous Proteins

HMMER, version 3.1b2 [53], was used to search the GAF (PF06018) and HTH (PF08222) domains of CodY in the deduced amino acid sequence derived from the bacterial genome and metagenome databases. The significance was set at an e-value < 10^−3^. A putative CodY was identified in proteins that contained both the GAF and HTH domains. BlastP analysis was used to calculate the average amino acid identity of the CodY-like proteins of Synergistetes (Table 1) and the CodY proteins of Firmicutes (Appendix A).

### 4.3. Phylogenetic Analysis of 16S rRNA and the CodY-like/CodY Proteins from Synergistetes and Other Phyla

The 16S rRNA genes from 35 strains of the phylum Synergistetes (Table 1), 17 species of the Firmicutes (Appendix A) and 12 species from the phyla Proteobacteria, Fusobacteria, Bacteroidetes, Actinobacteria and Aquificae (Appendix A) were retrieved from the database. An alignment of these 16S genes was performed using ClustalW integrated in MEGA version 7 [54] with standard parameters. The sequences of the *codY* genes from species of Synergistetes and Firmicutes were retrieved from the database, and an alignment of the deduced protein sequence was prepared using MUSCLE [55] integrated into MEGA version 7 with standard parameters. The maximum likelihood algorithm (ML) with 1000 bootstrap replications was used for phylogenetic tree reconstruction. The phylogenetic tree was managed and visualized by iTOL, version 5 [56].

### 4.4. Purification of the Recombinant CodY-like Protein

A preparation of the total cellular DNA of *C. evryensis* DSM19522 was purchased from the Leibniz Institute DSMZ (Braunschweig, Germany). The coding region of the putative *codY* (CLOEV_RS08630) was PCR amplified from chromosomal DNA by using the Blend Taq™-Plus (Osaka, Toyobo) with primers DSM19522_codY-SphI_S and DSM19522_codY-HindIII_AS. The estimated melting temperatures of primers DSM19522_codY-SphI_S and DSM19522_codY-HindIII_AS are 60.7 °C and 61.4 °C, respectively. The annealing temperature of the PCR was set at 56 °C (30 s), and reaction was performed for 35 cycles. Restriction recognition sequences were included in the primers to facilitate cloning. The PCR product was restriction digested and cloned into pQE30 (Qiagen, Hilden, Germany) in *E. coli* M15. The identity of the recombinant plasmid was verified by sequencing analysis. The recombinant histidine-tagged CodY-like protein (His-CodY_DSM_) was induced and purified from the recombinant *E. coli* strain by HIS-Select Nickel Affinity Gel (Sigma, St. Louis, MO, USA) under native conditions, according to the manufacturer’s recommendation. The concentration of the purified protein was determined by Bio-Rad Protein Assay based on the method of Bradford [57]. All primers used in this study are listed in Appendix A.

### 4.5. DAPA

All oligonucleotides used in DAPA were end-labeled using the Pierce biotin 3′ end DNA labeling kit (Thermo, Waltham, MA, USA). Two complementary biotin-labeled oligonucleotides containing the desired sequences were annealed and then used in DAPA reactions. Various amounts of purified His-CodY_DMS_ was mixed with 10 μL Mag-Beads- Ni-NTA (Tools, Taiwan) in separate reactions, with constant rotation for 30 min at room temperature. At the end of the reaction, the beads were washed with 1 mL reaction buffer (10 mM MgCl_2_, 50 mM KCl, 10 mM Tris, [pH 7.5], 1 mM EDTA, 50 µg ml^−1^ BSA) in a magnetic field to remove unbound proteins. The protein-DNA binding reaction was carried out in 50 μL of the reaction buffer containing 0.1 pmol biotin-labeled probe and poly[dI-dC] at 20 µg mL^−1^, and the reaction was incubated with tilt rotation for 30 min at room temperature. The final concentrations of His-CodY_DMS_ in the reactions were 0.02, 0.1, 0.5 and 2.5 μM, respectively. At the end of incubation, the beads were washed with 200 μL of reaction buffer three times, and then resuspended in 20 µL TE buffer (50 mM Tris, 10 mM EDTA [pH 8]) containing proteinase K at 2 mg mL^−1^. The digestion was carried out at 37 °C for 30 min to release the DNA. The final product was resolved on 6% non-denaturing polyacrylamide gels, electro-transferred onto nylon membranes, and detected using a chemiluminescent nucleic acid detection module kit (Pierce). The DAPA experiments were repeated six times with three batches of purified His-CodY_DMS_.

### 4.6. Binding Site Analysis and Target Gene Prediction

The sequence of a 400-bp region 5′ to the *xerC* gene, the first gene of the *codY*-containing operon of species of the phylum Synergistetes were extracted from the genomes of 35 cultivatable isolates, 17 human gut metagenome-assembled genomes and 22 ecological metagenome assembly sequences. All these sequences were then used for common motif discovery using MEME motif-searching algorithm [58]. Genera with less than three genomes available were excluded from the analysis. The default settings were used with the following exceptions: the minimum length was set to 15 bases and interrogated sequences with palindromes only.

To define the possible regulon of the CodY-like protein, all promoter regions of the 21 complete genomes in the phylum Synergistetes were scanned for 5′-AATTTTCWGAAAATT and the newly identified consensus binding sequence of the CodY-like protein by FIMO [59].

### 4.7. Functional Enrichment of the Identified Target Genes Regulated by the CodY-like Protein

The functional clustering of the target genes regulated by the CodY-like proteins was initially analyzed using Roary [60]. The identified target genes were further enriched based on predicted functions using EggNog [61], Clusters of Orthologous Groups (COGs) [62] and KEGG orthology [63]. The enrichment score was calculated using log_2_(N+1) (where N is the number of the identified genes). R script was used to generate the heat map showing the enrichment scores.

## Figures and Tables

**Figure 1 ijms-23-07911-f001:**
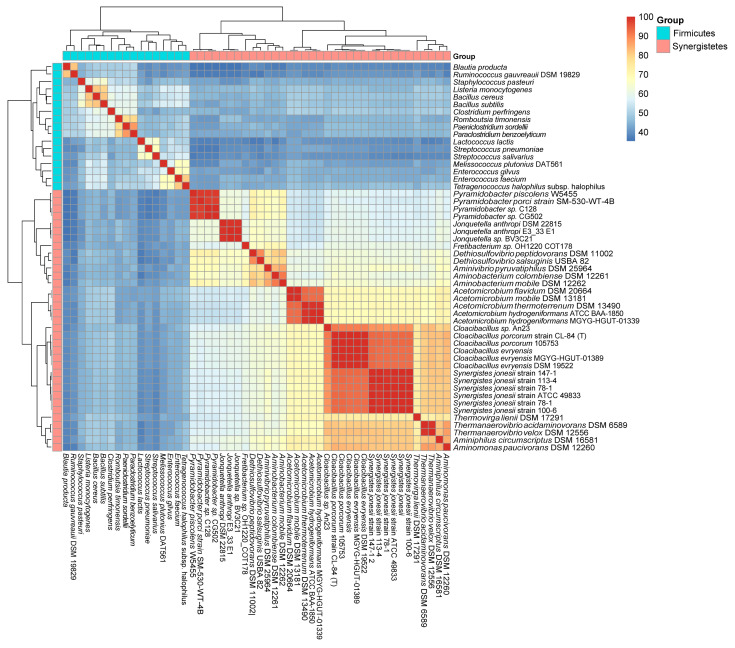
A heat map showing the identity between the CodY/CodY-like proteins from species of Firmicutes and Synergistetes. The score is the % identity of the deduced amino acid sequence of *codY* and its homologs.

**Figure 2 ijms-23-07911-f002:**
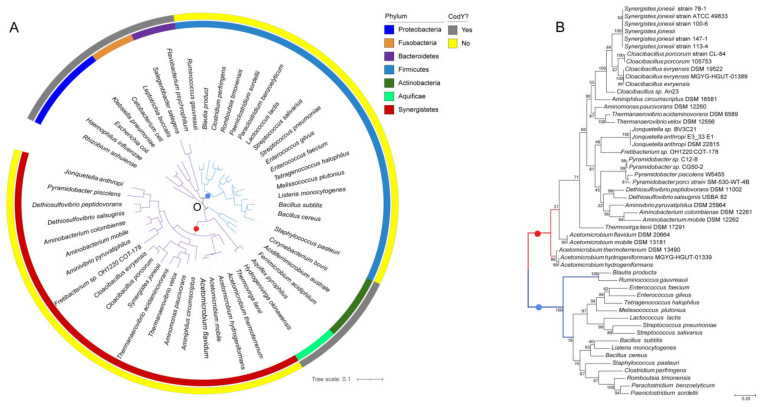
The phylogenetic relationship of species and the CodY/CodY-like proteins of Firmicutes and Synergistetes. (**A**) Phylogenetic tree of the 16S rRNA of representative species from various bacterial phyla. The presence (grey) and absence (yellow) of a *codY* gene in the chromosome of a species is shown on the outer circle. Species of different phyla are color-coded and shown in the inner circle. The phylum information is shown on the right. An ML distance tree based on the 16S rRNA sequences of 51 species from different phyla is shown inside the circle. Gram-negative bacteria are indicated by purple branches and Gram-positive bacteria are indicated by blue branches. (**B**) An ML distance tree based on 52 CodY homologous protein sequences of different species in the phyla Firmicutes and Synergistetes. The phylum Synergistetes is indicated by the red branch and the phylum Firmicutes is indicated by the blue branch. The numbers on the branches indicate the percentage of calculations performed by bootstrap.

**Figure 3 ijms-23-07911-f003:**
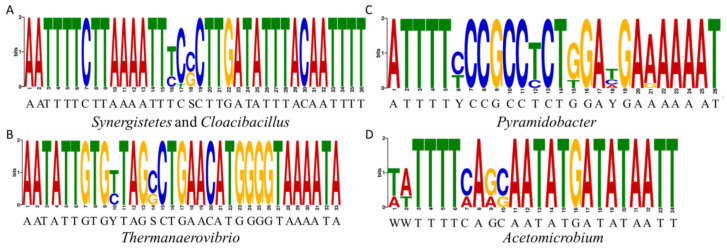
Motif logos representing the binding sites of the CodY-like proteins for five genera in the phylum Synergistetes. The logos were generated by the MEME function of the MEME suite. All 40 core binding sequences of the CodY-like protein listed in Appendix A were used to analyze the common motifs in *Synergistes* and *Cloacibacillus* (**A**), *Thermanaerovibrio* (**B**), *Pyramidobacter* (**C**) and *Acetomicrobium* (**D**).

**Figure 4 ijms-23-07911-f004:**
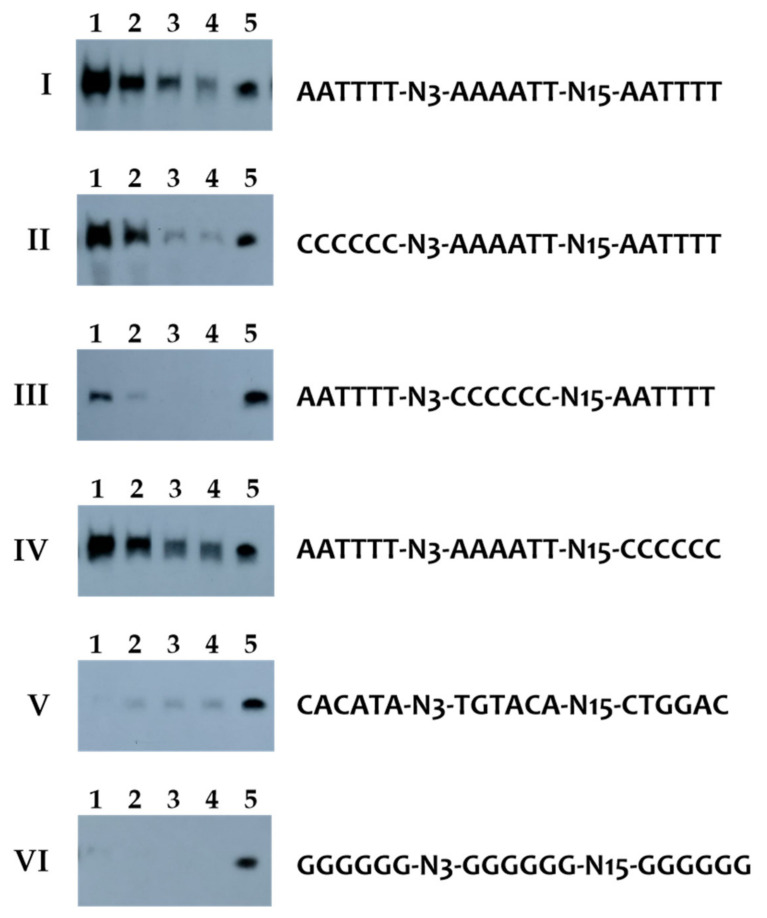
DAPA experiments showing the binding specificity of the CodY-like protein of *C. evryensis* DSM19522. Lanes 1 to 4, 2.5 μM, 0.5 μM, 0.1 μM, and 0.02 μM of the purified His-CodY_DSM_ were used in the binding reactions, respectively. Lane 5, 1.25% of the probe used in the reaction is shown to ensure a comparable number of labeled probes was used in each of the reactions. (I), the wild-type probe containing the predicted binding sequence 5′ to *xerC* of *C. evryensis* DSM1952 was used. (II), the 5′ AT-rich sequence of the probe is mutated. (III), the central AT-rich sequence of the probe is mutated. (IV), the 3′ AT-rich sequence of the probe is mutated. (V) and (VI), all three AT-rich sequences were mutated. The key features of the probes used in each reaction are listed right to the images. For clarity, only the sense strand of the probe is shown.

**Figure 5 ijms-23-07911-f005:**
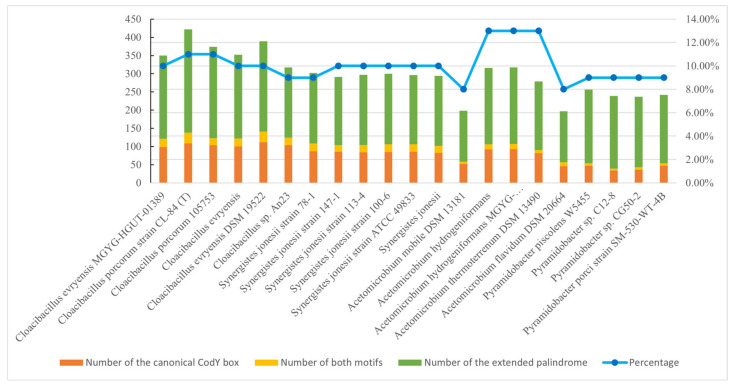
The distribution of genes regulated by the CodY-like protein in the available genomes of the phylum Synergistetes. The number of loci that harbor the canonical CodY box, the newly defined extended palindromic sequence, and both motifs in the promoter regions are shown in orange, green and light orange, respectively. The percentages of total targets in the genomes are indicated.

**Figure 6 ijms-23-07911-f006:**
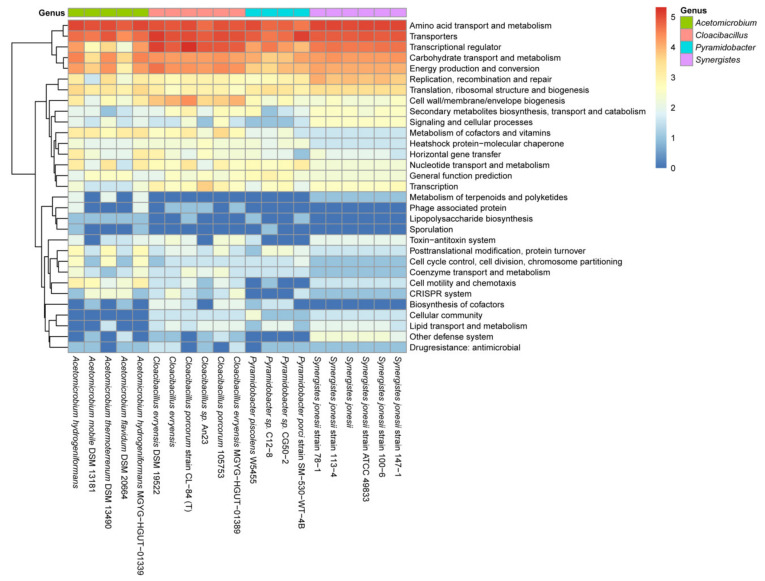
A pathway-enrichment heat map showing the clustering of loci regulated by the CodY-like protein. The positions of the four genera are color-coded and listed on the top of the figure. The number of loci regulated by the CodY-like protein was analyzed based on the predicted function. The number was converted using log_2_ (N + 1), where the N is the number of the identified target genes.

**Table 1 ijms-23-07911-t001:** The distribution of the putative CodY-like protein among the cultivable species of the phylum Synergistetes.

Locus_Tag	RefSeq	Assembly	GC (%)	Species
BUQ78_RS04225	NZ_FSQZ01000001	GCF_900129645	48.29	*Acetomicrobium flavidum* DSM 20664
HMPREF1705_RS06035	NZ_ACJX03000001	GCF_000160455	46.53	*Acetomicrobium hydrogeniformans*
FY315_RS06110	NZ_CABKOO010000001	GCF_902374045	46.53	*Acetomicrobium hydrogeniformans* MGYG-HGUT-01339
ANAMO_RS01275	NC_018024	GCF_000266925	47.94	*Acetomicrobium mobile* DSM 13181
BLU12_RS09355	NZ_FNPD01000014	GCF_900107215	48.46	*Acetomicrobium thermoterrenum* DSM 13490
K349_RS0111180	NZ_JAFY01000007	GCF_000526375	59.16	*Aminiphilus circumscriptus* DSM 16581
C8D99_RS00920	NZ_SORI01000001	GCF_000165795	67.58	*Aminivibrio pyruvatiphilus* DSM 25964
AMICO_RS07740	NC_014011	GCF_000025885	45.31	*Aminobacterium colombiense* DSM 12261
K360_RS0102980	NZ_JAFZ01000001	GCF_000526395	43.88	*Aminobacterium mobile* DSM 12262
APAU_RS04160	NZ_CM001022	GCF_000165795	67.58	*Aminomonas paucivorans* DSM 12260
FXY42_RS11030	NZ_CABKQM010000008	GCF_902374565	56.26	*Cloacibacillus evryensis* MGYG-HGUT-01389
CLOEV_RS08630	NZ_KK073872	GCF_000585335	55.95	*Cloacibacillus evryensis* DSM 19522
HMPREF1006_RS10935	NZ_JH414697	GCF_000238615	56.26	*Cloacibacillus evryensis*
BED41_RS09070	NZ_CP016757	GCF_001701045	54.84	*Cloacibacillus porcorum* strain CL-84 (T)
HF883_RS06600	NZ_JABAGT010000007	GCF_012844265	56.42	*Cloacibacillus porcorum* 105753
B5F39_RS01840	NZ_NFJQ01000002	GCF_002159945	55.63	*Cloacibacillus* sp. An23
DPEP_RS01550	NZ_ABTR02000001	GCF_000172975	54.41	*Dethiosulfovibrio peptidovorans* DSM 11002
B9Y55_RS09045	NZ_FXBB01000024	GCF_900177735	55.24	*Dethiosulfovibrio salsuginis* USBA 82
EII26_RS10880	NZ_RQYL01000028	GCF_003860125	62.36	*Fretibacterium* sp. OH1220_COT-178
JONANDRAFT_RS04180	NZ_CM001376	GCF_000237805	59.50	*Jonquetella anthropic* DSM 22815
GCWU000246_RS04765	NZ_GG697147	GCF_000161995	59.09	*Jonquetella anthropic* E3_33 E1
HMPREF1249_RS06160	NZ_AWWC01000024	GCF_000468895	59.31	*Jonquetella* sp. BV3C21
HMPREF7215_RS03875	NZ_ADFP01000047	GCF_000177335	58.15	*Pyramidobacter piscolens* W5455
B0D78_RS09560	NZ_MUHX01000026	GCF_002007215	57.20	*Pyramidobacter* sp. C12-8
D7D26_RS08945	NZ_RAWT01000042	GCF_003612005	58.62	*Pyramidobacter* sp. CG50-2
FYJ74_RS00750	NZ_VUNH01000001	GCF_009695745	58.90	*Pyramidobacter porci* strain SM-530-WT-4B
EH55_RS06015	NZ_JMKI01000026	GCF_000712295	55.86	*Synergistes jonesii* strain 78-1
JS79_RS06185	NZ_JPZR01000022	GCF_001757565	55.84	*Synergistes jonesii* strain 147-1
JS78_RS05700	NZ_JPZS01000024	GCF_001757415	55.84	*Synergistes jonesii* strain 113-4
JS77_RS05735	NZ_JPZQ01000030	GCF_001757495	55.85	*Synergistes jonesii* strain 100-6
JS73_RS05710	NZ_JQEL01000027	GCF_001757485	55.84	*Synergistes jonesii* strain ATCC 49833
JS72_RS09655	NZ_JQEK01000046	GCF_001757465	55.83	*Synergistes jonesii*
TACI_RS06905	NC_013522	GCF_000024905	63.79	*Thermanaerovibrio acidaminovorans* DSM 6589
THEVEDRAFT_RS02110	NZ_CM001377	GCF_000237825	58.78	*Thermanaerovibrio velox* DSM 12556
TLIE_RS02745	NC_016148	GCF_000233775	47.10	*Thermovirga lienii* DSM 17291

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
