# Peer review of "Identification of a Putative CodY Regulon in the Gram-Negative Phylum Synergistetes"

_ijms, 2022, doi:10.3390/ijms23147911_

Round 1
Reviewer 1 Report
In this work, the authors attempted to identify the putative CodY regulon in Gram-negative Synergistetes. Using various bioinformatics analyzes and the DNA affinity precipitation test (DAPA), this study confirmed the presence of CodY orthologs and their corresponding regulons in Gram-negative Synergistetes. CodY is the dominant regulator in Gram-positive Firmicutes and regulates various metabolic pathways and cellular processes. According to the authors, although Synergistetes and Firmicutes are phylogenetically distinct, both CodY and the CodY-like protein are conserved with respect to their regulatory function. Based on the potential targets of the CodY-like protein, the regulatory spectrum of the CodY-like protein overlaps with the Firmicutes CodY regulon. The authors conclude that, as in Firmicutes, regulation by a CodY-like protein is possibly necessary for Synergistetes microbes to deal with stress in their natural niches. The work is quite innovative and brings an interesting aspect to further understanding of the structure and function of Synergistetes strain.
Author Response
Thanks for the encouragement; we are quite excited to discover that CodY is not limited to Firmicutes, and the function of CodY is conserved across two distinct phyla. It will be interesting to find out the true effectors for the CodY-like protein in Synergistetes in the future studies.
Reviewer 2 Report
1. Line 445- please mention melting temperature and PCR cycle conditions used for respective PCRs.
2. Line 355- explain ecological niches mentioned here.
3. Line 403- discuss future options to evaluate effect of various stress situations on Cod-Y like protein expression and their application
Author Response
- Line 445:
The coding region of the putative codY (CLOEV_RS08630) of C. evryensis DSM19522 was PCR amplified from chromosomal DNA by using the Blend Taq™-Plus (Toyobo) with primers DSM19522_codY-SphI_S and DSM19522_codY-HindIII_AS. The estimated melting temperatures of primers DSM19522_codY-SphI_S and DSM19522_codY-HindIII_AS are 60.7oC and 61.4oC, respectively. The annealing temperature of the PCR was set at 56 oC, and the reaction was performed for 35 cycles. The above information is provided in the revised manuscript.
- Line 355:
Microbes in the phyla of Firmicutes and Synergistetes occupy a wide range of niches, and some of the niches are shared by both phyla, for instance, the oral cavity. However, Synergistetes are more commonly found in extreme environments whereas Firmicutes are not. To avoid confusion and overstatement, we have removed the last sentence of the paragraph in the revised manuscript.
- Line 403
As indicated in the manuscript, genetic studies in the microbes of Synergistetes are limited by the difficulties of cultivation and the lack of genetic systems. Before both obstacles are overcome, perhaps, in vivo chromatin immunoprecipitation (ChIP) assay coupled with quantitative PCR could provide an alternative approach to analyze the impact of the CodY-like protein in the expression of predicted target loci under various growth conditions in the cultivatable species. For instance, the impact of nutrient starvation on the binding of the CodY-like protein on sporulation-related genes.